# Differences in the Formation Mechanism of Giant Colonies in Two *Phaeocystis globosa* Strains

**DOI:** 10.3390/ijms21155393

**Published:** 2020-07-29

**Authors:** Dayong Liang, Xiaodong Wang, Yiping Huo, Yan Wang, Shaoshan Li

**Affiliations:** 1Key Laboratory of Ecology and Environmental Science in Guangdong Higher Education, School of Life Science, South China Normal University, Guangzhou 510631, China; liangdayong007@163.com; 2Research Center for Harmful Algae and Marine Biology, Jinan University, Guangzhou 510632, China; pouchetii@gmail.com (X.W.); epingfok@163.com (Y.H.)

**Keywords:** *Phaeocystis globosa*, harmful algal bloom, giant colonies formation, transcriptome, metabolic pathway

## Abstract

*Phaeocystis globosa* has become one of the primary causes of harmful algal bloom in coastal areas of southern China in recent years, and it poses a serious threat to the marine environment and other activities depending upon on it (e.g., aquaculture, cooling system of power plants), especially in the Beibu Gulf. We found colonies of *P. globosa* collected form Guangxi (China) were much larger than those obtained from Shantou cultured in lab. To better understand the causes of giant colonies formation, colonial cells collected from *P. globosa* GX strain (GX-C) and ST strain (ST-C) were separated by filtration. Morphological observations, phylogenetic analyses, rapid light-response curves, fatty acid profiling and transcriptome analyses of two type cells were performed in the laboratory. Although no differences in morphology and 18S rRNA sequences of these cells were observed, the colonies of GX strain (4.7 mm) are 30 times larger than those produced by the ST strain (300 μm). The rapid light-response curve of GX-C was greater than that of ST-C, consistent with the upregulated photosynthetic system, while the fatty acid content of GX-C was lower than that of ST-C, also consistent with the downregulated synthesis of fatty acids and the upregulated degradation of fatty acids. In summary, the increased energy generated by GX-C is allocated to promote the secretion of extracellular polysaccharides for colony formation. We performed a physiological and molecular assessment of the differences between the GX-C and ST-C strains, providing insights into the mechanisms of giant colonies formation in *P. globosa*.

## 1. Introduction

The genus *Phaeocystis* (Prymnesiophyceae) is globally distributed in the marine environment, ranging from tropical to polar oceans, and plays important roles in the global carbon and sulfur cycles and in regional food webs [1]. Among the six species to have been identified, *P. globosa* is a common cause of harmful algal blooms that often produces red tides by forming colonies in temperate and trophic zones [2,3].

*P. globosa* has been reported along the coast since the 1990s in China [4]. During the period 2001–2007, blooms of *P. globosa* were frequently reported throughout the year, including extensive areas from the Bohai Sea (the North China Sea) to the coast of the Hainan Province (the South China Sea), strongly impacting on local fisheries [5,6]. From 2010-2019, outbreaks of *P. globosa* not only affected the local aquaculture and coastal environment, but also posed a potential threat to nuclear power cooling systems in coastal areas. Thus, blooms of *P. globosa* have become an important environmental problem requiring urgent mitigation.

*P. globosa* has a complex life cycle involving the formation of colonies with an exopolysaccharide matrix tegument and a variety types of solitary cells [7,8,9]. The solitary cells are generally 3–10 μm in diameter, while colonies can reach up to 3 cm in diameter [4,7]. However, under laboratory culture, the largest colonies only reach a few hundred microns diameter, if they form colonies at all. The competitive advantage of *P. globosa* over other phytoplankton species is primarily due this ability to form colonies. In fact, colony formation is one of the remarkable plastic phenotypic characteristics of *Phaeocystis* species [10]. The large size and tough outer layer of *Phaeocystis* colonies inhibit their ingestion by zooplankton, particularly small-sized grazers, thus colony formation and enlargement are considered to be defense strategies that reduce predation risk [10,11,12]. However, the mechanisms associated with *P. globosa* colony formation remain unclear. We found colonies of *P. globosa* collected form Guangxi (China) were much larger than those obtained from Shantou (China) cultured in lab. The great difference in colony size between the two strains makes them the ideal materials of the giant colony formation mechanism study in the life cycle of *P. globosa*. 

Interestingly, we observed that the colonies of *P. globosa* (GX) were much larger than *P. globosa* (ST) cultured in lab, although it has been difficult to distinguish subspecies in *P. globosa* with respect to their phylogeny and morphology. This provides us with a good opportunity to study the mechanism and metabolic pathway of the formation of giant colonies. In this study, the morphology of *P. globosa* was characterized by light and electronic microscopies. In addition, we performed a phylogenetic analysis using 18S ribosomal RNA (rRNA) gene to further characterize *P. globosa*. Physiological differences between two strains were also observed by performing rapid light-response curve (RLC) and fatty acid profiling analyses, while an Illumina sequencing platform was used to determine the transcriptional profiles of the two strains. The results revealed differences between colonial cells of the two strains with respect to the metabolic pathways associated with photosynthesis, carbon source allocation and exopolysaccharide secretion. The results of this study can contribute to an understanding of the cause of giant colony formation in *P. globosa* and provide important reference for understanding the molecular regulatory mechanism of colony formation.

## 2. Results

### 2.1. Morphological Observation and Phylogenetic Analysis 

The colony was broken to release colonial cells prior to LM and SEM observations. The average diameter of each viable colonial cell was approximately 6 μm for the two strains, which decreased by 50% to only 2–3 μm under SEM, with colonial cells generally appearing as round-shaped with two plastids (Figure 1a,c). SEM observations showed that the flagella and a short haptonema are replaced by two short appendages (0.15–0.45 μm), inserted into the center of the apical pole between the two plastids, and these have a smooth surface without scales (Figure 1b,e). Colonial cells are irregularly inlayed at the periphery of colony (Figure 1c,f).

The phylogenetic tree derived by the ML method is illustrated in Figure 2. The blast result shows an identity percentage of 99.88% and 2 nucleotides (nt) differ between ST and GX strains. Similar 18S rRNA phylogenies were obtained from the ML and BI analysis. The phylogenetic tree was inferred by the ML method with bootstrap values indicated for ML/BI. The phylogeny shows that ST and GX strains belong to the same sub-group within the *P. globosa* clade. Hongkong2, the strain also isolated from the coastal water of South China Sea, belongs to the same sub-group with ST and GX. The other sub-group of the *P. globosa* clade contains five strains isolated from coastal water of Europe. These results suggest that the two study strains are closely related and cannot be well distinguished in morphology and phylogeny.

### 2.2. Growth of Solitary Cells and Colony Formation

The abundance of solitary cells for *P. globosa* ST was significantly higher than that observed for *P. globosa* GX (*p* < 0.05), with the highest ST cell abundance being approximately 8000 cells mL^−1^ (Figure 3a). The number of colonies formed by GX was also lower than that of ST, but the diameter of the GX colonies was much larger than that of observed for the ST colonies (Figure 3b). For example, the average diameter of ST colony is only approximately 300 μm, while that of GX can reach 1.3 cm, spanning two orders of magnitude (*p* < 0.05; Figure 3c,d). This result suggests that there is a great difference in life history between the two strains, where ST is dominated by solitary cells, while GX is dominated by large colonies with almost no solitary cells.

### 2.3. RLCs, PS II Electron Transport Rates and Fatty Acid Profiling

Rapid light-response curves (RLCs) and fatty acid contents were measured for the two strains of colonial cells (ST-C and GX-C). After being cultured for 10 days, no significant differences in the α parameter (initial slope of RLC) or point of light saturation (*I*_k_) were observed between ST-C and GX-C (Figure 4, *p* > 0.05, ANOVA). The maximal relative electron transport rate (*P*_m_, rETR_max_) in the GX-C group was significantly increased (20%) compared to that observed for ST-C (Figure 4, *p* < 0.05, ANOVA,). The RLCs of GX-C were higher than those observed for ST-C, indicating that ST-C had higher photosynthetic activity (Figure 4).

The fatty acid methyl ester (FAME) contents and profiles were further analyzed on the 10th day, and compared between ST-C and GX-C (Table 1). Eight fatty acids were detected in *P. globosa*, including three saturated fatty acids (SFAs), two mono-unsaturated fatty acids (MUFAs) and three polyunsaturated fatty acids (PUFAs). The fatty acid profiles indicated that the fatty acid components of *P. globosa* were C14, C16 and C18, with SFAs accounting for 71–76% of total fatty acids. ST-C also contained MUFAs and PUFAs, similar to GX, but unsaturated fatty acids were not detected in latter strain except for C24:1. Thus, the fatty acid content of GX-C is evidently lower than that of ST-C, which accounted for only 15% of ST-C (Table 1, *p* < 0.05, ANOVA). This result suggests that although GX-C has a greater photosynthetic capacity, the obtained energy is not used to form energy storage materials such as fatty acids.

### 2.4. Transcriptome Analysis

To study the differences in the colonial cells at the transcriptome level, the cultured cells (10 days) were subjected to RNA extraction and transcriptome sequencing analysis. Three biological replicates for GX-C and ST-C were used to ensure statistical comparability and the reliability of the data. We set ST as the control group, and the change of gene is the difference between GX and ST in the transcriptome. The raw data ranged from 25,608,632 to 31,698,971 reads per sample. After producing more than 24 million clean reads, low-quality and adapter sequences were removed. Subsequently, 38,570 genes were identified as being differentially expressed at significant levels, including 17,402 upregulated genes and 21,162 downregulated genes (Figure 5a). We enriched for differentially expressed genes with Gene Ontology (GO) and observed no significant differences in cellular composition and molecular function. The genetic differences between the two strains were primarily concentrated in biological processes, such as single-organism, metabolic, cellular metabolic processes (Figure 5b).

#### 2.4.1. Enhanced Photosynthesis

The expression of photosystem I (PS I) and photosystem II (PS II) in GX-C did not change significantly relative to that observed for ST-C. Most of the light harvesting center proteins, such as the Light harvest complex (LHCI and LHCII), were upregulated in both PS I and PS II (Figure 6a). During plant photosynthesis, the photons absorbed by LHCII are transferred to the RC via LHCI, with ATP subsequently produced and energy released through a series of electron transfers. The upregulated expression of light absorption protein suggests that GX-C has a greater light absorption capacity, which is consistent with the higher RLCs observed for this strain. Upregulation of the electron transporter plastocyanin also suggested that the photosynthesis apparatuses may be enhanced, but the expression of genes encoding chloroplast ferredoxins (FD, 25135.0; 38557.0) and ferredoxin-NADP^+^ reductase (FNR, 34545.1; 62284.1) were also downregulated. Ferroredoxin is the last electron carrier that receives the electrons generated by the photostimulated light system and transfers them to FNR to produce NADPH in the process of noncyclic photophosphorylation. Thus, downregulation of the gene encoding FNR suggests a decrease in NADPH levels.

#### 2.4.2. Enhanced Fatty Acid Degradation and Suppressed Fatty Acid Biosynthesis

Suppressed fatty acid biosynthesis and enhanced fatty acid degradation were observed in GX-C (Figure 6b). The expression of acetyl-coenzyme carboxylase (*accA*, *accC* and *accD*), an important regulatory enzyme in the fatty acid synthesis, was significantly downregulated. Most notably among these genes, the *accD* gene (41388.39699) was downregulated 20-fold compared to GX (Figure 6b). The fatty acid synthase (FAS) of plants functions within fatty acid synthase system II, including β-ketoacyl-ACP synthase (FabH), β-ketoacyl-ACP reductase (FabG), β-hydroxyacyl-ACP dehydratase (FabZ) and enoyl-ACP reductase (FabI). Unlike the mammalian type I system, which uses a large multifunctional enzyme, the type II system uses a series of monofunctional proteins that each catalyze one step in the biosynthesis pathway [13]. Another important enzyme in fatty acid synthesis was also generally downregulated in our study. Genes coding for nearly all the enzymes throughout the fatty acid degradation pathway were consistently upregulated, including acyl-CoA synthetase (ACS), acyl-CoA dehydrogenase (ACADM) and acetyl-CoA acyltransferase 1 (ACAA1). ACS homologous proteins comprise the long-chain acyl-CoA synthetase (ACSL, 46738.21248) and bubblegum (ACSBG, 41388.21236) subfamilies that activate long-chain and very-long-chain fatty acids to form acyl-CoAs (Figure 6b). Suppressed fatty acid biosynthesis and enhanced fatty acid degradation leads to a sharp decrease in fatty acid content, which is consistent with our physiological data.

#### 2.4.3. Increased Exopolysaccharide Biosynthesis

The synthesis of exopolysaccharide can be divided into three stages. First, sucrose is converted to Glc-1P and Man-6P, and many genes encoding the enzymes responsible for these reactions are upregulated, such as PGM (76567.0, 7557.0) during these processes, which isomerizes Glc-6P to Glc-1P, and HK (73956.0) also take parts in the biosynthesis of Glc-6P. UDP-glucose, a sugar donor of glyconucleotides, is synthesized in vivo by UDP-sugar pyrophosphorylase (USP, 41388.15749). Second, UDP-Gal is converted to NDP-sugar, and Man-1P is indirectly converted to GDP-Man. Genes encoding GALE (41388.35035, 41388.12882) were significantly upregulated, but those encoding GMDS (45241.0) and GMPP (32270.0, 33011.0) were downregulated during this process. Based on UDP-Glc and guanosine diphosphate mannose (GDP-Man), other NDP sugars are further converted through the action of NDP-sugar interconversion enzymes [14]. Finally, various NDP-sugars form growing polysaccharide chains by the action of glycosyltransferases (GTs) that were notably upregulated, such as ALG7 (73869.0), ALG13 (71877.1), ALG1 (75759.1), and ALG2 (41388.21383).

## 3. Methods

### 3.1. Cultures and Culturing Conditions

The pure culture of *P. globosa* GX strain (collected in Guangxi 2017) and ST strain (collected in Shantou 2003) were conserved in the Algae species preservation room of the Research Center for Harmful Algal and Marine Biology of Jinan University. Cultures were maintained at 20 °C, 12:12 h light:dark cycle with PAR approximately 80–100 µmols^−1^ m^−2^. Solitary and colonial cells were separated by filtering the stock culture through a 20 μm sieve. The cells were centrifuged (5000 rpm, 10 min) and then frozen stored at −80 °C for RNA extraction.

### 3.2. Morphological Observation by Using of Light Microscopy and Scanning Electron Microscopy

Both strains were evaluated under an inverted light microscope (LM, Olympus-CKX41, Tokyo, Japan) in late exponential growth stage. Algae were centrifuged, and the pelleted cells were washed with 0.1 mol/L of phosphate buffer (pH 7.2). Subsequently, the cells were fixed with glutaraldehyde (2.5% final concentration) and refrigerated at 4 °C for 10 h. Alga were transferred to a holder containing a filter membrane (1 μm), and the samples were dehydrated with ethanol (through the sequence 20%, 40%, 50%, 60%, 80%, 90%, 100%). Subsequently, the filter was dried, a dehydrating agent was used to eliminate residual moisture, and the cells were then coated with gold and observed under a scanning electron microscope (SEM, Zeiss SIGMA 500, Oberkochen, Germany). Sixty cells were randomly selected to calculate the average diameter under light microscope, and 30 cells were examined under SEM.

### 3.3. DNA Extraction and Phylogenetic Analyses

DNA of *P. globosa* was extracted by using a plant DNA Extraction Kit (Takara, Japan) and PCR amplification using the primers l8SF (5′-CCTGGTTGATCCTGCCAG-3′) and l8SR 5′-(TTGATCCTTCTGCAGGTTCA-3′) [15], and repeated three times. The PCR thermocycling conditions involved 95 °C for 5 min; 35 cycles at 94 °C for 45 s; 55 °C for 45 s; 72 °C for 90 s; and finally at 72 °C for 10 min.

The PCR products were purified by PCR MasterMix kit (Sangon, China) and sent to Sangon Biotechnology Co., Ltd. for sequencing. After manual proofreading, the sequence was searched using BLAST in GenBank. Additional 18S rRNA sequences representing *P. globosa* from different regions were retrieved from GenBank for phylogenetic analysis. *Emiliania hyxleyi* (Noelaerhabdaceae, Prymnesiophyceae) was used as the out-group in Bayesian reconstruction using MrBayes 3.2 with the selected substitution model (SYM+G) [16]. Maximum likelihood (ML) were analyzed on the T-REX web server with GTR+G [17]. 

### 3.4. Algal Growth Analysis

Solitary cells of ST and GX strains were individually maintained in Erlenmeyer flasks in triplicate containing fresh prepared f/2 medium. The experiments were initiated with density of 10^3^ cells ml^−1^ [18]. During the experiment, the conical flasks were shaken daily to evenly distribute the *P. globosa* cells and reduce their adherence to the flask wall. All incubation lasted for 10 days. Samples were withdrawn every 2 days and fixed with Lugol’s solution. The abundance of solitary cells, colony diameters and colonies numbers were determined using a counting chamber under an inverted microscope (LM, Olympus-CKX41, Tokyo, Japan). 

### 3.5. Rapid Light-Response Curve (RLC) and Fatty Acid Profiling Analyses

To measure the photosynthetic efficiency of two strains algae, chlorophyll fluorescence measurements were performed via chlorophyll fluorescence meter (PAM-2100, Walz, Germany) connected to an excitation-detection unit (ED-101US /M, Walz, Germany). Curve fitting was performed using Statistica based on the formula described by Platt (1982) (*P* = *P_m_* (1 − e ^− α×PAR/ Pm^) e ^− β×PAR/ Ps^) [19]. *P* is the relative electron transfer rate; *P*_m_ is the maximum potential electron transfer rate without photoinhibition; α is the initial slope of the *P*-*I* curve; and β is the light suppression parameter.

A modified Bligh Dyer (m-BD) method (1959) was used to extract total fatty acids from colonial cells [20]. The algae powder (0.2g) was added mixed solvent (10 mL, chloroform: methanol: distilled water = 1:2:0.8), 40 kHz ultrasonic treatment for 5 min, 5,000× g centrifugation for 5 min, and the supernatant was collected. The sodium chloride (1 mL, 5%) was added, mixed and set aside. The total fatty acids in the chloroform layer were obtained by rotary evaporator. Fatty acid was tested by gas chromatography mass spectrometry (Agilent Technologies, Palo Alto, CA, USA). Heptadecanoic acid (C17:0, Sigma, USA) was used as a standard to determine the fatty acid, and analysis was performed as described in Lu (2014) [13]. The lipid unsaturation (DLU) was calculated in the light of the equation: (DLU(▽/mole) = [1.0 × (% monoene) + 2.0 × (% diene) + 3.0 × (% triene)]/100) [21].

### 3.6. RNA Exaction, Data Filtering and Mapping

For RNA exaction, samples were resuspended in 2 mL of RNA extraction buffer (1:1 mix of aqua-phenol and buffer L and then incubated with DNase I (Takara, Japan) for 30 min at 37 °C to remove genomic DNA. The RNA quality analysis, library construction, sequencing, data filtering and mapping were performed by Novogene Bioinformatics Technology Co., Ltd. (Beijing, China). The data discussed in this publication have been deposited in the NCBI Gene Expression Omnibus and are accessible through GEO accession number GSE141662 (https://www.ncbi.nlm.nih.gov/geo/query/acc.cgi?acc=GSE141662).

### 3.7. Quantification of Gene Expression Levels and Differential Expression Analysis

Differential expression analysis of GX and ST colonial cells was performed using samples with biological replicates with the DESeq R package (1.10.1). DESeq provides statistical routines for determining differential expression from digital gene expression data using a model based on the negative binomial distribution. The resulting *p*-values were adjusted using the Benjamini and Hochberg (1995) approach to control the false discovery rate [22,23]. Differentially expressed genes (DEGs) were identified with DESeq using the criteria |log_2_ fold change| > 2 and *p*-value < 0.05. The functional classification of DEGs was performed according to gene ontology (GO) annotations, and the pathway analysis was carried out according to KEGG.

### 3.8. Statistical Analysis

Data were calculated from three replicates per alga strain (GX and ST) and are presented as means ± standard error. Significant differences between solitary and colonial cells were determined using one-way analysis of variance (ANOVA) with SPSS 22.0 version (IBM, Armonk, NY, USA), and statistical significance was set at a *p*-value < 0.05 compared with the control (Dong et al., 2019).

## 4. Discussion

The primary reason for the frequent outbreak of *P. globosa* blooms on the southeastern coast of China is the formation of colonies [10]. Thus, studying the mechanism of *P. globosa* colony formation has become important. Although no differences in morphology and 18S rRNA sequences of these cells were observed, the colonies of GX strain are 30 times larger than those produced by the ST strain. The 18S rRNA gene is potentially conserved also. When compared to other rRNA genes such as 28S rRNA gene or the ITS, Medlin (2007) observed that ITS1 exhibits substantial inter- and intraspecific sequence divergence and provided more resolution among strains [15]. However, strains assayed in this study differ greatly in life history, with GX primarily behaving as macrocolonies, while ST primarily behaves as solitary cells. Different life histories may be caused by different environmental conditions. Colony formation can be influenced by many factors, such as nutrients, zooplankton feeding, and temperature [24,25]. The macrocolonies of *P. globosa* strains may be due to the heteromorphic life cycle of this species, in which haploid and diploid cells can reproduce sexually [26,27], since the process of sexual reproduction can lead to gene exchange and fusion between different cells. Many studies have found that colonies of *P.*
*g**lobosa* become smaller after several months when cultured in the lab [28], but the GX strain can still form large colonies (maximum 2 cm) when cultured in the lab even after 2 years. The differences between *P. globosa* strains with respect to their colony formation may provide a breakthrough in the study and understanding of the mechanism of *P. globosa* colony formation. By comparing the transcriptomes of the two strains, we can assess the differences in their metabolic pathways and identify the key genes affecting the formation of colonies.

The relative electron transport rate (rETR) measured by fluorescence reflects not only the photosynthetic capacity of cells, but also the *P*-*I* curve, which is an effective alternative to traditional methods. We observed that GX-C has a greater photosynthetic ability than ST-C, meaning that GX-C can fix more carbon. The transcriptome results showed that *LHC* gene expression was significantly upregulated in GX-C, indicating enhanced light absorption. LHC contains three transmembrane helices, and the binding sites for chlorophyll a, chlorophyll b and lutein, and LHC can be divided into two subclasses: LHCI and LHCII [29]. The absence of an *Lhca* gene can cause the loss of all LHC I proteins, thus reducing the absorption and transmission of light energy in *Chlamydomonas* [30]. Andersson (2003) showed that the chlorophyll content of *Arabidopsis thaliana* decreased with the absence gene of *LHCB1* and *LHCB2* [31]. Although the light saturation point did not change, the photosynthetic efficiency decreased by 10–15%. Interestingly, the genes encoding components for the last portion of the photosynthetic electron transport were downregulated, including ferredoxin, FNR and ADP/ATP transporters. Previous studies on diatoms have shown that nitrogen deficiency leads to an inhibition of photosynthetic proteins, including FNR [32]. The decrease in ferredoxin levels may be due to the decrease of nitrogen content caused by continuous culturing. Except for the energy transferred by ferredoxin, ATP and NADP can be generated by glycolysis, the TCA cycle and activated oxidative phosphorylation. This phenomenon of inconsistent regulation of photosynthetic system has also been observed in a study of lipid accumulation in *Coccomyxa subellipsoidea* C-169 under high CO_2_ conditions, which is believed to be a unique mechanism for algae to adapt to rapid growth and lipid accumulation [33]. Enhancement of photosynthesis provides guarantees for maintaining rapid growth and releasing exopolysaccharides in large colonies even if some photosynthetic genes are downregulated. Despite the greater photosynthetic capability of GX-C, we observed that its fatty acid contents were actually lower. The genes encoding ACC (*accA*, *accC* and *accD*), an important enzyme in fatty acid synthesis, were shown to be significantly downregulated. The ACC enzyme in plants has been heavily investigated, and the overproduction of ACC in *Brassica napus* L has been shown to increase the lipid content by 6% [34]. Wang (2018) showed that the upregulation of ACC in proteome of the oleaginous microalga *Auxenochlorella protothecoides* led to an increase in fatty acid production under high-temperature conditions [35]. Interestingly, if only fatty acid synthesis pathways were downregulated, the algae sometimes did not have lower fatty acid levels. We also observed upregulation of the fatty acid metabolic pathway of GX-C, including the levels gene code of ACS, ACADM, ACAA1 and MFP2. The acyl-CoA synthetase (ACS) is a key enzyme in fatty acid decomposition. After knocking out the acyl carrier protein synthase in *Synechococcus* sp. and *Synechococcus elongatus*, an enzyme that is highly homologous to ACS, Kaczmarzyk (2010) showed that the levels of free fatty acids in cells increased significantly [36]. It is widely assumed that any physiological advantage must have a trade-off in metabolic costs [37]. The fatty acid synthesis pathway was significantly downregulated, and the fatty acid decomposition pathway was significantly upregulated in GX-C, suggesting that the content of fatty acids was lower in the cells forming large colonies. *P. globosa* can form large colonies, and the mucilaginous matrix can account for more than 50% of the total carbon in a colony [38]. The distribution of limited energy to promote the formation of colonies rather than the synthesis of energy storage materials may be an effective survival strategy.

The success of *P. globosa* in the environment is primarily attributed to its ability to form colonies. Hamm (1999) observed that the primary components of the outer colony are polysaccharides that are tough, elastic, transparent and have large pores on the surface [39]. Polysaccharide production pathways are conserved across prokaryotes and eukaryotes, with monosaccharides being converted to nucleotide sugars that are assembled into polysaccharides by the activity of glycosyltransferases (GTs) [40,41]. We observed that the polysaccharide synthesis pathway genes of GX strain were generally upregulated compared with ST strain and attempted to identify the key genes responsible for controlling exopolysaccharide secretion, which are key to colony formation. Glucokinase (GK) can regulate the storage rate of sugar and free sugar levels in plants, as well as the metabolic rate of glycolysis and the pentose oxidative phosphoric acid pathway, and it plays a key role in the distribution of carbon flow in plants [42]. Interestingly, carbohydrates have been identified as signaling molecules, similar to plant hormones that participate in sugar signal transduction in plants and play a strong role in regulating the synthesis of fatty acids and other metabolites [43]. These findings indicate that the upregulation of the GK gene will not only affect the synthesis of polysaccharides, but also regulate the synthesis of fatty acids. Plant polysaccharides are formed by the active nucleotide-diphospho-sugar (NDP-sugar) precursors, which are added to the residues of polysaccharides and glycoconjugates by the activity of various glycosyltransferases [14,44]. We observed that the *ALG* gene was also upregulated, primarily in the Golgi, and catalyzed the transfer of N-acetylglucosamine residues from UDP-N-acetylglucosamine to D-mannose at the core of the receptor N-sugar chain. ALG is an oligosaccharide linked to an asparagine residue side chain amide nitrogen in the protein peptide chain that is essential for the effective secretion of plant glycoproteins [45]. In addition, we found that the genes encoding GMPP and GMDS were downregulated. GMPP is an important enzyme in both eukaryotic and prokaryotic cells because it catalyzes the formation of GDP-Man from Man-1-P [46]. GMD is a member of the NDP-sugar modifying subfamily of the short-chain dehydrogenases/reductases [47]. Downregulation of genes encoding GMPP and GMD resulted in the decrease of extracellular polysaccharide synthesis by GDP-Man. This suggests that the synthesis pathway of EPS is different between the two colonial cells, and the secretion of EPS in GX is mainly from glucose and galactose.

We know that UDP-sugars are an important source of nucleotide sugar based on previous studies [48,49]. UDP-sugar-producing pyrophosphorylases are important enzymes in the process of glucose metabolism, and are responsible for the synthesis and metabolism of UDP sugar in organisms [50,51]. There are three types of UDP glycopyrophosphorylase, including uridine diphosphate glucose pyrophosphorylase (UGPase), uridine diphosphate acetylglucosamine pyrophosphorylase (UAPase) and uridine diphosphate sucrose pyrophosphorylase (USPase) [52]. The *USP* gene of colonial cells (GX) is upregulated compared with colonial cells (ST) according to transcriptomic date. Aslam (2018) showed that with the decrease in temperature, the exopolysaccharide secretion of the diatom *Fragilariopsis cylindrus* increased, and the level of *USP* was significantly upregulated in its proteome [40]. These results are also consistent with our previous comparisons of colonial and noncolonial cells (unpublished date), indicating that upregulation of *USP* gene can make cells release more extracellular polysaccharides and promote the formation of colonies, and the verification of *PgUSP* function will be a focus of our future work.

## 5. Conclusions

In this study, we attempted to combine physiological and transcriptome sequencing analyses to assess differences in colony size among different *P. globosa* strains (GX and ST). We showed that GX colonies were much larger than those produced by ST, although the two strains could not be distinguished based on molecular and morphological data. Thus, exploring the differences in their metabolic processes will allow us to better understand the mechanism of giant colonies formation. Possible working model for the colony formation in two *P. globosa* strains was established (Figure 7). GX-C has greater metabolic potential, including enhanced photosynthesis, suppressed fatty acid biosynthesis and enhanced fatty acid degradation as well as increased exopolysaccharide biosynthesis, which are consistent with the physiological indicators. More energy was absorbed by photosynthesis for exopolysaccharide biosynthesis and secretion, which may promote the formation of the giant colony in *P. globosa*. In addition, our findings suggest that *PgUSP* is a key gene in promoting ESP secretion in *P. globosa*, since its upregulation is consistent with our previous studies. Overall, this study provides a comprehensive understanding of the mechanism of forming giant colonies in *P. globosa*, a valuable source for future inhibition of harmful algal blooms.

## Figures and Tables

**Figure 1 ijms-21-05393-f001:**
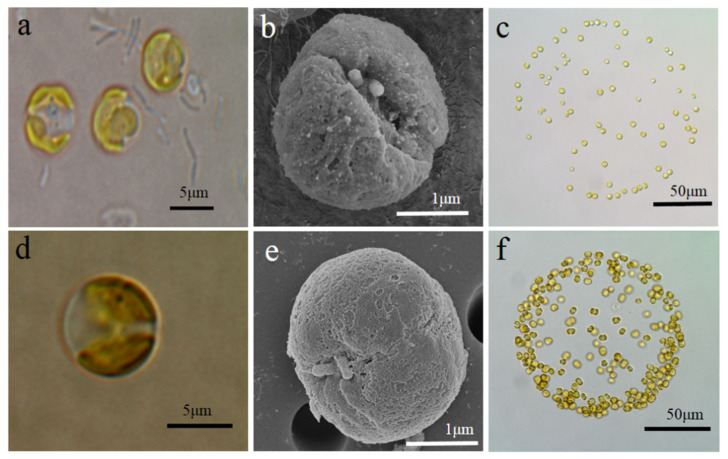
LM and SEM photographs of colonial cells and colonies. ST colonial cell showing the two short appendages inserted near the center of the apical pole and a colony (200 μm) embodying 60 colonial cells (**a**–**c**); GX colonial cell showing the two short appendages and a colony (150 μm) embodying 140 colonial cells (**d**–**f**).

**Figure 2 ijms-21-05393-f002:**
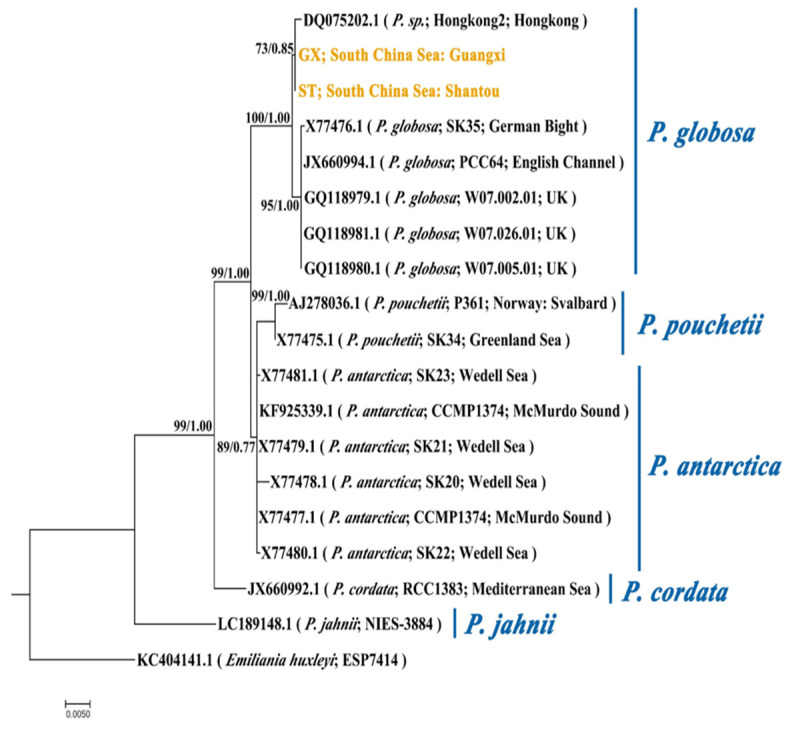
Phylogeny of *Phaeocystis* based on 18S rRNA sequences (1646 characters after data processing). The phylogenetic tree was inferred by the maximum likelihood (ML) method. Numbers at nodes represent the bootstrap values for the ML bootstrap analysis and Bayesian posterior probabilities. The strains in yellow are objects in this study. The scale bar indicates the number of nucleotide substitutions per site. ST represents the *P**. globosa* collected in Shantou and GX represents the *P. globosa* collected in Guangxi.

**Figure 3 ijms-21-05393-f003:**
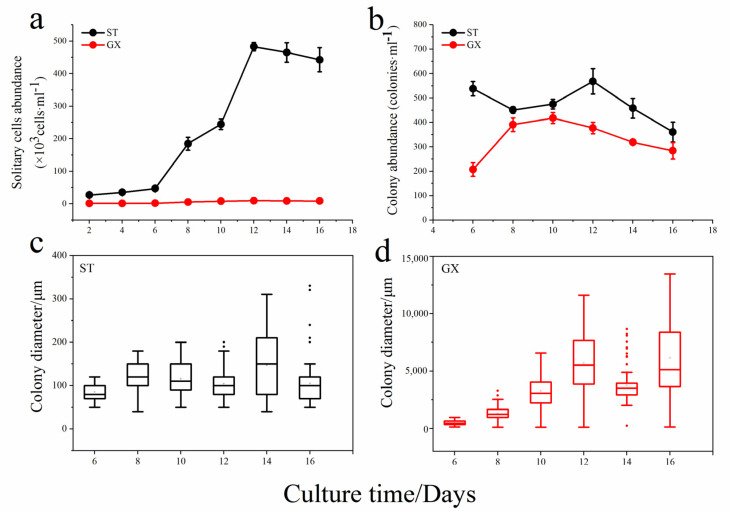
Growth curves of the two *P. globosa* strains. Abundances of solitary cells (**a**); colony abundance (**b**); colony diameter of ST (**c**); colony diameter of GX (**d**). ST represents *P. globos**a* originally collected in Shantou and GX represents *P. globosa* originally collected in Guangxi.

**Figure 4 ijms-21-05393-f004:**
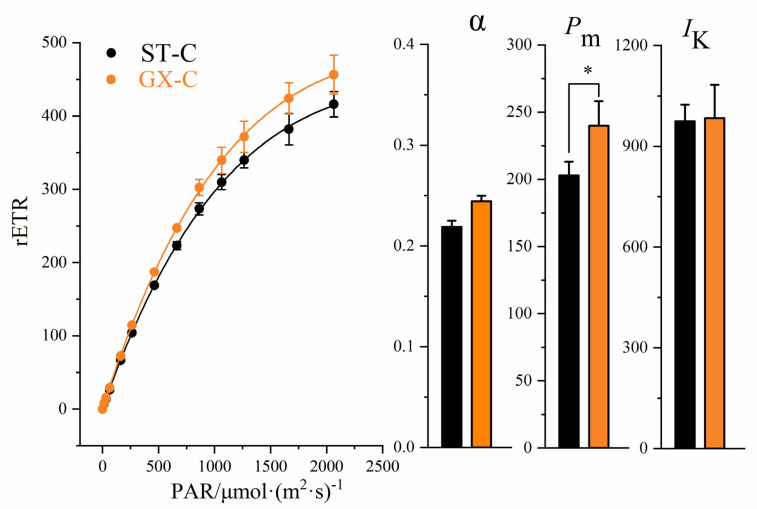
Comparison of the rapid light-response curves, the initial slope of RLC (α), the maximal relative electron transport rate (*P_m_*) and the point of light saturation (*I_k_)* between ST-C and GX-C. ST-C represents the colonial cell of the *P. globosa* ST strain (black) and GX-C represents the colonial cell of the *P. globosa* GX strain (orange). * represents a statistically significant difference of *p* < 0.05.

**Figure 5 ijms-21-05393-f005:**
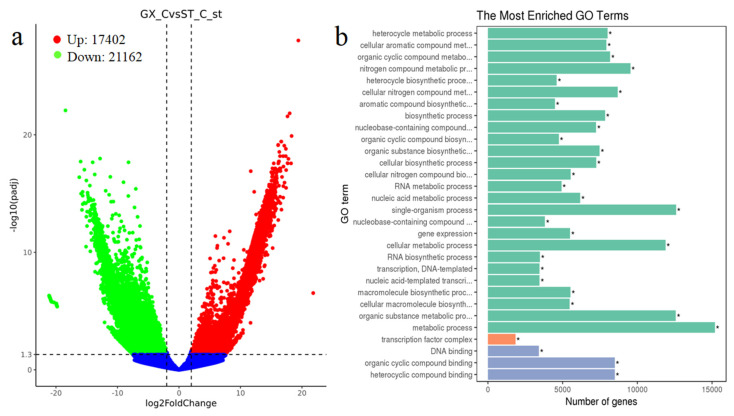
Diagram of the *P. globosa* mRNA sequence analysis from colonial cells (GX-C) and colonial cells (ST-C). (**a**) Volcano map of different genes in two *P. globosa* strains (green represents downregulated genes; red represents upregulated genes; blue represents genes with no significant difference). (**b**) GO terms of different genes enrichment. X axis represents number of genes, Y axis represents GO term name (green represents biological process; orange represents cellular component; blue represents molecular function). * represents a statistically significant difference of *p* < 0.05.

**Figure 6 ijms-21-05393-f006:**
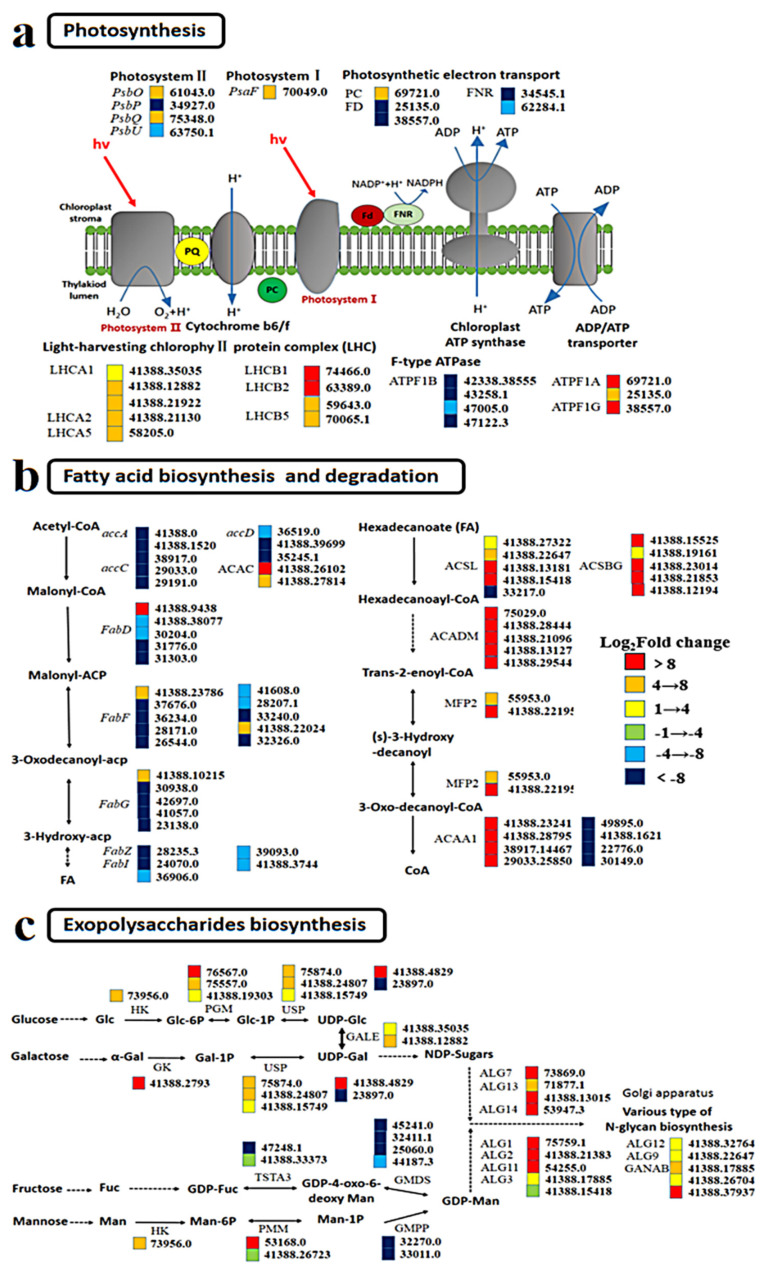
Responses to changes between ST and GX in the mRNA abundance of genes involved in metabolic pathways and biological processes. (**a**) Photosynthesis; (**b**) fatty acid biosynthesis and degradation; (**c**) exopolysaccharide biosynthesis. Key enzymes are included in the map and presented as their names, gene IDs and fold changes as indicated by colored boxes (red, orange and yellow represent upregulated; green, sky blue and dark blue represent downregulated). Chemical compound abbreviations: LHC, light-harvesting complex chlorophyll a/b binding protein; PC, plastocyanin; FD, ferredoxin; FNR, ferredoxin-NADP^+^ reductase; ATPF, F-type transporting ATPase; ACACA, acetyl-CoA carboxylase; ACSL, long-chain acyl-CoA synthetase; ACSBG, bubblegum; MFP2, multifunctional protein; ACAA1, acetyl-CoA acyltransferase 1; HK, hexokinase; PGM, phosphoglucomutase; USP, UDP-sugar pyrophosphorylase; GALE, UDP-glucose 4-epimerase; GK, galactokinase; PMM, phosphomannomutase; GMPP, mannose-1-phosphate guanylyltransferase; GMDS, GDP-mannose 4,6-dehydratase; TSTA3, GDP-L-fucose synthase; ALG, acetylglucosaminyltransferase; GANAB, mannosyl-oligosaccharide alpha-1,3-glucosidase.

**Figure 7 ijms-21-05393-f007:**
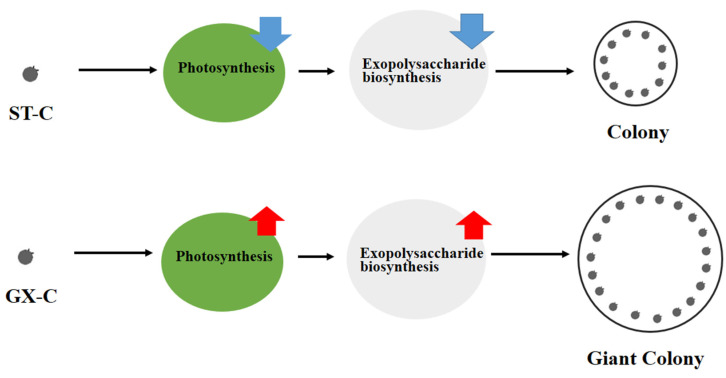
Possible working model for the colony formation in two *P. globosa* strains. ST-C represents the lab colonial cells of *P. globos**a* originally collected in Shantou and GX-C represents the lab colonial cells of *P. globosa* originally collected in Guangxi; the red arrow means upregulation of metabolic process and the blue represents down regulation of metabolic process.

**Table 1 ijms-21-05393-t001:** Fatty acid profiles for different *P. globosa* cells (mg/g). ST-C represents the colonial cells of the Shantou strain and GX-C represents the colonial cells of the Guangxi strain. *ND*—not detected.

Fatty Acids	GX-C	ST-C	*p*-Value
C14:0	0.156 ± 0.017	2.851 ± 0.453	
C16:0	0.397 ± 0.003	1.151 ± 0.091	*p* < 0.05
C18:0	0.196 ± 0.012	0.390 ± 0.017	*p* < 0.05
C18:1	ND	0.690 ± 0.215	*p* < 0.05
C18:3ω6	ND	0.283 ± 0.010	
C18:3ω3	ND	0.292 ± 0.090	
C22:6ω3	ND	0.772 ± 0.092	
C24:1	0.375 ± 0.003	ND	
Total	1.124 ± 0.012	6.429 ± 0.075	*p* < 0.05

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
