# Peer review of "Differences in the Formation Mechanism of Giant Colonies in Two Phaeocystis globosa Strains"

_ijms, 2020, doi:10.3390/ijms21155393_

Round 1
Reviewer 1 Report
Liang et al investigated and reported that the formation mechanism of giant colonies in two Phaeocystis globosa strains.
The research is well designed and each data is clear and can be agreed.
However, before a publication, the authors must improve and consider some points shown below.
1) One of the main conclusions in this article is that the large amount of secreted extracellular polysaccharides for colony formation.
If so, it seems to be not so difficult to measure the content of extracellular polysaccharides.
2) In some parts of this article, the authors' expressions are overinterpreted. They need to be tone-down if there is no direct evidence.
3) A summarized and/or a possible model as a new Figure will be useful for a better understanding of this study.
4) strain name:
The authors must add an explanation about "GX algal strain (GX-C)" and "ST algal strain (ST-C)".
Further, GX and ST are abbreviations of some words?
"-C" means what?
5)Lines68-75:
The logical expansion in these sentences is too bad.
6) The titles of the result part are not informative.
The titles must include information about what's the critical findings/conclusion, etc.
7) Fatty acid analysis:
The fatty acids investigated in this study are "total fatty acids isolated from the cells"? They must clear this point, also must indicate the method of preparation of fatty acids from the cells detailed.
8) Figure legends:
The figure legends do not have enough information for explaining each Figure, especially, Figures 4, 5, and 6.
The authors must add informative information in each figure legend.
9) The explanation for the section of transcriptome analysis:
This section's explanation is not sufficient for the readers.
The authors must explain kindy and carefully the data.
Reviewer 2 Report
Manuscript ID: ijms-850614
Title: Differences in the formation mechanism of giant colonies in two Phaeocystis globosa strains
Major comments:
Abstract is very long and need to concise with experimental methods.
Introduction, Materials and methods sections written and discussed well. Also Results and Discussion section organized very well.
However, Figures 5 and 6 need be replaced with with high resolutions figure because quality is very poor.
Conclusion sections need to revised and key results need to be highlighted.
Round 2
Reviewer 2 Report
Thanks to the authors for addressing all the comments.